Proceedings Track

# Adaptive Safety in Reinforcement Learning via Advanced Lagrangian Optimization

**Feihong Zhang**[1]     **Tianyi Zhang**[1]     **Maanping Shao**[2]
[1]**College of Artificial Intelligence/Vehicle and Mobility, Tsinghua University**
[2]**Institute for Interdisciplinary Information Sciences, Tsinghua University**

## 1. Background

### 1.1. Introduction

Reinforcement Learning (RL) is a key area in machine learning, achieving success in robotics, autonomous driving, and games Li (2023). However, real-world applications require Safe RL to prevent harmful behaviors by integrating safety directly into learning Ray et al. (2019).

### 1.2. Importance of the Problem

Traditional RL focuses on maximizing rewards without considering safety during learning Achiam et al. (2017). Safe RL methods often use Lagrangian approaches to enforce constraints but face issues with instability and slow convergence Munos et al. (2016).

### 1.3. Impact of the Proposed Solution

The proposal aims to develop an optimization algorithm that improves Safe RL by better handling constraints during training. This aims to achieve faster convergence and better safety compliance, enabling RL agents in safety-critical environments.

## 2. Problem Definition

### 2.1. Mathematical Formulation

We consider a constrained Markov Decision Process (CMDP) Puterman (2014) defined as:

$$\mathcal{M} = (\mathcal{S}, \mathcal{A}, P, r, c, \gamma),$$

Objective:

$$\max_{\pi} \quad J_r(\pi) = \mathbb{E}\left[\sum_{t=0}^{\infty} \gamma^t r(s_t, a_t)\right],$$

$$\text{subject to} \quad J_c(\pi) = \mathbb{E}\left[\sum_{t=0}^{\infty} \gamma^t c(s_t, a_t)\right] \le d,$$

The meaning of each letter in the formula is explained in the Appendix Table 1.

## 3. Related Work

### 3.1. Existing Approaches

Constrained Policy Optimization (CPO), Lagrangian-Based Methods like RCPO, and PID Lagrangian Methods are three approaches to incorporating constraints into policy optimization in reinforcement learning.

CPO Achiam et al. (2017) uses trust-region methods to ensure updates meet safety constraints, RCPO Tessler et al. (2018) integrates constraints via Lagrange multipliers, and PID Lagrangian Methods Stooke et al. (2020) apply PID control to update multipliers.

### 3.2. Gaps in Existing Work

**Stability Issues:** Current methods can struggle with stable enforcement of safety constraints during learning Achiam et al. (2017).
**Convergence Speed:** Slow convergence hampers practical applicability in real-world scenarios Berkenkamp et al. (2017).
**Complexity:** Some algorithms require complex computations, limiting scalability Bastani et al. (2018).

## 4. Proposed Method

To enhance stability and improve learning efficiency, we focus on reducing instability in constraint handling. We've developed an enhanced Lagrangian algorithm to better balance the gradients of the objective and constraint functions during iterations, which is key to improving overall performance 1.(The algorithm pseudocode is listed in the Appendix.)

### 4.1. Advanced Lagrangian Optimization Algorithm

Due to page limitations, we present the pseudo-code of the algorithm designed so farBoyd and Vandenberghe (2004).

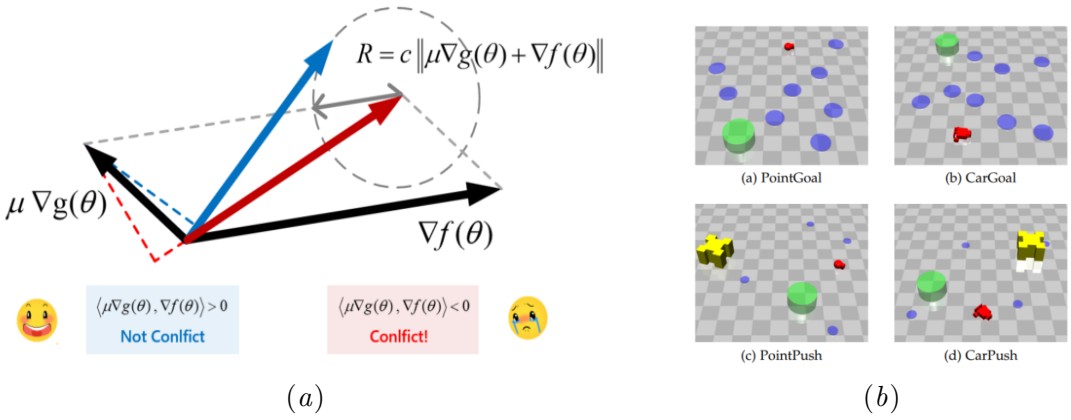

Figure 1: (a)Gradient update mechanism of Advanced Lagrangian Optimization Algorithm(The meanings of the letters in the figure are listed in Appendix Table 2); (b)Snapshots of four Safety Gym tasks.

### 4.2. Datasets and Experimental Setup

**Environments:** Safety Gym. Provides environments with various safety constraints. Tasks include navigating to goals while avoiding hazards.
**Baseline Approaches:**Yang et al. (2023) Qin et al. (2024) We will compare our method with:
(1)Constrained Policy Optimization (CPO),
(2)PID Lagrangian Methods,
(3)Standard Lagrangian Methods.

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

## Appendix A. Symbol meaning

Table 1: CMDP process symbol meaning

| Symbol | Meaning |
| --- | --- |
| $\mathcal{S}$ | State space |
| $\mathcal{A}$ | Action space |
| $P(s'\|s,a)$ | Transition probability |
| $r(s,a)$ | Reward function |
| $c(s,a)$ | Cost function (safety constraints) |
| $\gamma$ | Discount factor |
| $\pi_\theta(a\|s)$ | Policy function |

Table 2: ALO Algorithm symbol meanning(part)

| Symbol | Meaning |
| --- | --- |
| $\nabla f(\theta)$ | Gradient of the objective function |
| $\nabla g(\theta)$ | Gradient of the constraint function |
| $\mu$ | Lagrange multiplier |
| $c$ | An adaptive hyperparameter |
| $R$ | The size of the search radius |

## Appendix B. Algorithmic pseudo-code

---

**Algorithm 1:** Advanced Lagrangian Optimization Algorithm

---

1. **Initialize:**

   - Hyperparameter: $c_0 \in (0, 1)$, Lagrange multiplier $\mu_0 \geq 0$ (initially 0)
   - Learning rates: $\eta > 0$, $\beta > 0$, Scaling factor $\gamma \in (0, 1)$
   - Policy parameter: $\theta_0$

2. Set $k = 0$

3. /* Compute Gradients */
   $$g_1^k = \nabla_\theta f(\theta_k), \; g_2^k = \nabla_\theta g(\theta_k)$$

4. /* Update Lagrange Multiplier */
   $$\mu_{k+1} = \max\left(0, \mu_k + \beta \, g(\theta_k)\right)$$

5. /* Compute Fusion Gradient */
   $$g_0^k = g_1^k + \mu_{k+1} \, g_2^k$$

6. /* Optimize $\alpha$ and Compute Update Direction */

   (a) $\alpha_k^* = \arg\min_{\alpha \in [0,1]} \left(\langle g_\alpha^k, \, g_0^k \rangle + c_k \, \|g_0^k\| \, \|g_\alpha^k\|\right)$

   (b) $d_k = g_0^k + \frac{1}{2\lambda_k} \, g_{\alpha^*}^k$

7. /* Update Policy Parameter */
   $$\theta_{k+1} = \theta_k - \eta \, d_k$$

8. /* Update Hyperparameters */
   $$c_{k+1} = \gamma \, \sin\left(\frac{\theta_k}{2}\right)$$

9. Increment $k \leftarrow k + 1$

---

