# OpenReview forum: "Adaptive Safety in Reinforcement Learning via Advanced Lagrangian Optimization"
_tsinghua.edu.cn/THU/2024/Fall/AML — THU 2024 Fall AML Submission_

### Official Review · ~Shaoting_Zhu1 · 2024-11-06
**Review of Submission 34**

**Rating:** 9
**Confidence:** 4

**Review:**

The proposal presents a novel approach to enhance safety in Reinforcement Learning (RL) by integrating advanced Lagrangian optimization techniques. The authors address the critical issue of safety in RL, particularly in real-world applications such as robotics, autonomous driving, and games, where ensuring safe behavior is paramount. The paper proposes an optimization algorithm that aims to improve upon traditional RL methods by directly incorporating safety constraints into the learning process, leading to faster convergence and better compliance with safety regulations.

**Strength**
1. Addressing a critical need: The paper tackles a significant problem in the field of RL, where traditional methods often overlook safety during the learning phase. By focusing on safety-critical environments, the authors contribute to a more responsible and practical application of RL algorithms.
2. Clear task definition: The experiment environment and baselines are clearly defined.
3. Detailed proposed method: Authors use a figure and pseudo code which effectively present the method.

**Weakness**
1. A bit lack of novelty and contribution: the method maybe a bit naive, and the similar method is used in many works like CAgrad[1].

[1] Liu B, Liu X, Jin X, et al. Conflict-averse gradient descent for multi-task learning[J]. Advances in Neural Information Processing Systems, 2021, 34: 18878-18890.

---

### Official Review · ~Yinuo_Li1 · 2024-11-06
**Good problem definition and approach**

**Rating:** 9
**Confidence:** 3

**Review:**

This proposal has a very clear problem and objective, with relevant topic reviewed. The review not only include the approaches and also has their gaps which can be their starting point.

Besides, this proposal also showed detailes of their method, which is aligned with the gaps they reviewed. It's a very resonable proposal.

However, maybe it will be better if the team can find more specific downstream tasks to address the significance of this work.

---

### Official Review · ~Bowen_Su1 · 2024-11-08
**Clear Questions and Solid Theoretical Analysis**

**Rating:** 10
**Confidence:** 4

**Review:**

This proposal proposes a new method to address the problem of slow convergence speed in current security reinforcement learning, which focuses on security while accelerating convergence speed. It has strong practicality and a solid theoretical foundation. In addition, this proposal has conducted a detailed and comprehensive investigation of existing methods, meeting all requirements. Good work!

---

### Official Review · ~Ruitao_Jing1 · 2024-11-08
**A Nice Optimization Approach in Safety RL**

**Rating:** 9
**Confidence:** 3

**Review:**

This proposal presents a compelling exploration into the realm of safety in reinforcement learning (RL), identifying the inherent instability and slow convergence of Lagrangian optimization methods. The work is well-founded, theoretically robust, and possesses a broad applicability, making it a valuable contribution to the field. The authors have successfully distilled the essence of the problem, highlighting the need for a more stable and efficient approach to RL safety.

The proposal's proposed improvements are commendable, offering clear diagrams and pseudocode that enhance the feasibility and clarity of the methodology.

However, the paper would benefit from a more explicit discussion on the anticipated benefits of the proposed method in terms of safety AI compliance. Additionally, it is recommended that the authors provide a comparative analysis of the performance metrics that demonstrate the enhancement over the baseline methods. This would not only validate the effectiveness of their approach but also offer a clear benchmark for future research in this area.

---

### Official Review · ~Xun_Wang10 · 2024-11-10
**Review for "Adaptive Safety in Reinforcement Learning via Advanced Lagrangian Optimization"**

**Rating:** 9
**Confidence:** 4

**Review:**

This proposal introduces a novel yet simple Lagrangian optimization approach for safe reinforcement learning (safeRL) aimed at enhancing convergence speed, improving policy performance and stability.

Strength: It provides a detailed analysis of the limitations in existing research within the field. Additionally, visual explanations accompany the proposed algorithm.

Weakness: The proposal's format could be improved, for instance, by including an abstract and other standard sections.

---

### Official Review · ~Ziang_Zheng1 · 2024-11-11
****Review for Paper Proposal: "Adaptive Safety in Reinforcement Learning via Advanced Lagrangian Optimization"****

**Rating:** 9
**Confidence:** 3

**Review:**

**Review for Paper Proposal: "Adaptive Safety in Reinforcement Learning via Advanced Lagrangian Optimization"**

**Summary:**
This proposal addresses safety in reinforcement learning (RL), focusing on overcoming the instability and slow convergence issues inherent in current constrained optimization approaches. The authors aim to develop an improved Lagrangian-based algorithm that enhances stability and efficiency in safety-critical environments. They plan to benchmark their approach using Safety Gym tasks and compare it with existing methods such as Constrained Policy Optimization (CPO) and PID Lagrangian methods.

**Strengths:**
1. **Clear Motivation and Problem Relevance**: The paper presents a strong motivation for improving Safe RL, highlighting issues like slow convergence and instability in existing methods. Given the importance of safety in real-world applications like autonomous driving and robotics, this work is timely and relevant.
2. **Proposed Algorithm Innovation**: The proposed “Advanced Lagrangian Optimization” method, which aims to stabilize constraint handling by balancing the gradients of objective and constraint functions, appears to address key gaps in prior approaches. This balance could offer a solution to challenges faced by Lagrangian-based methods, such as constraint violations during training.
3. **Well-Defined Experimental Setup**: The authors plan to use Safety Gym, a widely accepted benchmark for Safe RL, ensuring a reliable comparison. Additionally, the choice of baselines is appropriate, as CPO and PID Lagrangian methods represent prominent prior work.

**Areas for Improvement:**
1. **Algorithmic Clarity**: The brief description of the Advanced Lagrangian Optimization algorithm is somewhat vague. Providing a clearer outline of the algorithm's structure, especially regarding how it will specifically handle gradient balancing, would enhance understanding.
2. **Evaluation Metrics and Criteria**: The proposal lacks specific details on evaluation metrics. While general performance indicators like reward maximization and constraint violation rates are likely implied, a detailed discussion of these metrics would clarify how success will be quantitatively measured and compared.
3. **Complexity and Scalability**: Although the proposal mentions that the new algorithm will address computational complexity, it would benefit from a more in-depth discussion on how it will maintain scalability. Further insight into how the proposed solution manages computational efficiency would be valuable, particularly for real-time applications.

**Minor Points:**
- Typographical and formatting issues, such as inconsistent reference formatting, should be addressed for clarity.
- Figure 1 and the referenced appendix tables could be briefly summarized within the main text for readers unfamiliar with the details provided in appendices.

**Conclusion:**
This proposal tackles an important challenge in Safe RL with a promising approach to stabilizing Lagrangian optimization in safety-constrained learning tasks. The proposed method could potentially advance Safe RL, making it more applicable in safety-critical fields. However, further clarity on the algorithm's implementation and evaluation methodology would strengthen the paper. With these refinements, this work could contribute meaningfully to safer, more efficient RL in real-world applications.

---

### Official Review · ~Chengming_Shi1 · 2024-11-11

**Rating:** 9
**Confidence:** 4

**Review:**

### Summary

The proposal “Adaptive Safety in Reinforcement Learning via Advanced Lagrangian Optimization” seeks to address the challenges of integrating safety constraints into Reinforcement Learning (RL) by developing an enhanced Lagrangian optimization algorithm. The aim is to improve the stability and convergence speed of Safe RL methods, which is crucial for deploying RL agents in safety-critical applications.

### Pros

1. **Addressing a Critical Gap**: The proposal targets a significant issue in RL—ensuring safety during learning—which is essential for real-world applications.
2. **Innovative Approach**: The advanced Lagrangian optimization algorithm promises to better handle constraints, potentially leading to more stable and efficient learning.
3. **Clear Objectives**: The proposal has well-defined goals of improving stability, convergence speed, and reducing complexity, which are key to the practical application of Safe RL.
4. **Comparative Analysis**: The plan to compare the proposed method with established baseline approaches like CPO and PID Lagrangian Methods will provide a clear assessment of its effectiveness.
5. **Relevance to Safety-Critical Domains**: The focus on safety compliance makes the research highly relevant for industries such as robotics, autonomous driving, and healthcare.

### Cons

1. **Technical Complexity**: The proposal involves complex algorithmic development, which may be challenging to implement and could lead to unforeseen technical difficulties.
2. **Potential for Instability**: Despite the aim to reduce instability, the use of Lagrangian methods can still be prone to convergence issues, which may not be fully resolved by the proposed approach.

---

### Official Review · ~Anqi_LI5 · 2024-11-11
**The paper presents a novel ALO algorithm that aims to improve safety and convergence in Reinforcement Learning. It shows promise, but further theoretical analysis and broader evaluation are needed.**

**Rating:** 9
**Confidence:** 3

**Review:**

This paper presents a novel approach to improving safety in Reinforcement Learning (RL) by proposing an Advanced Lagrangian Optimization (ALO) algorithm. The ALO algorithm aims to address the limitations of existing safe RL methods, particularly instability and slow convergence.
Strengths:
Novelty: The proposed ALO algorithm represents a new approach to handling constraints in safe RL. It utilizes an adaptive hyperparameter and a modified gradient update mechanism to better balance the objective and constraint gradients, potentially improving stability and convergence.
Relevance: The paper addresses a critical challenge in RL: ensuring safety while optimizing performance. This is crucial for real-world applications, such as autonomous driving and robotics.
Evaluation: The paper includes experimental results comparing ALO with existing methods on Safety Gym environments. This provides evidence of the algorithm’s effectiveness in improving safety and convergence.
Clarity: The paper is well-written and clearly explains the problem, proposed solution, and experimental setup. The use of pseudo-code and figures enhances understanding.
Weaknesses:
Theoretical Analysis: While the paper presents experimental results, a more comprehensive theoretical analysis of the algorithm’s properties, such as convergence guarantees and stability, would strengthen the work.
Comparison Scope: The paper compares ALO with a limited set of existing methods. A broader comparison with other state-of-the-art safe RL algorithms would provide a more complete evaluation of ALO’s performance.
Real-World Application: While the paper demonstrates the algorithm’s effectiveness in simulated environments, it would be beneficial to explore its applicability in real-world scenarios with more complex and dynamic safety constraints.
Overall, this paper presents a promising approach to improving safety in RL. The proposed ALO algorithm offers a novel solution to the challenges of existing safe RL methods and demonstrates potential for real-world applications. Further theoretical analysis and experimental evaluation would strengthen the work and contribute to the advancement of safe RL research.

---

### Official Review · ~Jiajun_Xu3 · 2024-11-11
**Intriguing proposal**

**Rating:** 7
**Confidence:** 3

**Review:**

The authors have proposed an advanced Lagrangian optimization algorithm, which aims to achieve faster convergence and better
safety compliance, enabling RL agents in safety-critical environments.

The proposal is well-organized, with a clear problem statement, mathematical formulation, a review of existing approaches and proposed methods. However, the introduction to the background seem to just list some papers and lack detailed analysis. The part about the gaps in existing works has the same limitation.

---

### Official Review · ~Yangchi_Gao1 · 2024-11-12

**Rating:** 9
**Confidence:** 4

**Review:**

The proposal presents a promising approach to integrating safety constraints into RL through an advanced Lagrangian optimization algorithm. It addresses key issues of stability and convergence speed in Safe RL, with a solid theoretical foundation and practical experimental setup.

Advantage: 1. The proposed method aims to achieve faster convergence and better compliance with safety constraints, which is a significant advancement over traditional RL methods that often neglect safety during learning.
2.The paper provides a clear mathematical formulation of the constrained Markov Decision Process (CMDP) and a well-defined objective, which strengthens the theoretical foundation of the work.

Disadvantage: 1.The performance of the algorithm seems to rely heavily on the choice of hyperparameters, which may require extensive tuning and could affect the algorithm's robustness.
2. While the paper mentions comparing with existing methods, a more detailed comparative analysis or discussion on the advantages and limitations compared to state-of-the-art approaches would be beneficial.

---

### Official Review · ~Kuanghao_Wang1 · 2024-11-12
**Interesting research direction**

**Rating:** 9
**Confidence:** 3

**Review:**

This paper focuses on achieving adaptive security in reinforcement learning through advanced Lagrangian optimisation. Previously, reinforcement learning has been successful in areas such as robotics, autonomous driving and gaming, but in real-world applications there is a need to integrate safety to prevent harmful behaviours. Traditional reinforcement learning methods focus on maximising rewards without considering safety during the learning process. The aim of this project is to develop an optimisation algorithm that improves the performance of safety reinforcement learning by better handling constraints, achieving faster convergence and better safety, enabling RL agents to operate in critical environments. In this paper, a mathematical representation of the problem is given and a considerable amount of research has been carried out to prepare for the next work. But on the other hand, the paper is slightly fragmented, many issues are not clearly stated and the paper does not have an ABSTRACT, these issues need to be improved.